# Zoonotic Episodes of Scabies: A Global Overview

**DOI:** 10.3390/pathogens11020213

**Published:** 2022-02-06

**Authors:** Barbara Moroni, Luca Rossi, Charlotte Bernigaud, Jacques Guillot

**Affiliations:** 1Department of Veterinary Sciences, University of Turin, Largo Paolo Braccini 2, 10095 Grugliasco, Italy; luca.rossi@unito.it; 2Research Group Dynamic, Ecole Nationale Vétérinaire d’Alfort, UPEC, USC Anses, 94704 Maisons-Alfort, France; bernigaud.charlotte@gmail.com (C.B.); jacques.guillot@oniris-nantes.fr (J.G.); 3Department of Dermatology, AP-HP, Hôpital Henri Mondor, Faculté de Santé, UPEC, 94000 Créteil, France; 4Department of Dermatology, Parasitology and Mycology, Ecole Nationale Vétérinaire de Nantes, Oniris, 44307 Nantes, France

**Keywords:** pseudoscabies, zoonotic scabies, sarcoptic mange, zoonosis, animals, *Sarcoptes scabiei*

## Abstract

Zoonotic scabies (ZS), also referred to as “pseudoscabies”, is considered a self-limiting disease with a short incubation period and transient clinical skin signs. It is commonly thought that *Sarcoptes scabiei* mites from animals are unable to successfully reproduce and persist on human skin; however, several ZS case reports have mentioned the persistence of symptoms and occasionally mites for weeks. The aim of this review was to collect and organize the sparse literature explicitly referring to *S. scabiei* zoonotic transmission, focusing on the source of the outbreak, the circumstances leading to the transmission of the parasite, the diagnosis including the identification of the *Sarcoptes* “strain” involved, and the applied treatments. A total of 46 articles, one conference abstract and a book were collected describing ZS cases associated with twenty animal hosts in five continents. Dogs were by far the most common source among pet owners, while diverse livestock and wildlife contributed to the caseload as an occupational disease. Genetic epidemiological studies of ZS outbreaks are still limited in number, but tools are available to fill this knowledge gap in the near future. Further research is also needed to understand the apparent heterogeneity in the morbidity, disease severity and timing of the response to treatment among people infected with different animal-derived strains.

## 1. Introduction

Scabies, also named sarcoptic mange when referring to animals, is a contagious parasitic skin disease caused by the burrowing mite, *Sarcoptes scabiei* (DeGeer, 1778) (Sarcoptiformes: Sarcoptidae), affecting more than 150 mammal species worldwide. It has been stated that “no other permanent parasitic mite has such a large variety of hosts as does *S. scabiei*” [1]. Scabies was listed by the World Health Organization (WHO) among the neglected tropical diseases in 2017, acknowledging the need for greater awareness on the part of practitioners and health organizations and for a global consensus on control guidelines and strategies. According to the WHO, more than 200 million people suffer from scabies globally, but the real extent of the disease is likely grossly underestimated [2,3]. The source of human scabies is commonly attributed to direct or, less frequently, indirect (via fomites) contact with affected people. More rarely, the origin of scabies episodes is traced back to contact with affected animals. Zoonotic scabies (from here on, ZS), also referred to as “pseudoscabies”, is considered a self-limiting disease, with a short incubation period and transient clinical skin signs. Usually, ZS is limited to some topographic regions of the body (i.e., chest, abdomen, hands), resolving with avoidance and/or treatment of the animal source [4]. It is commonly thought that animal variants of *S. scabiei* are unable to successfully reproduce and persist on human skin [5]. However, experimental human infections with scabies mites of dog origin resulted, in four of six volunteers, in a successful replication with hatching and development of mites [6,7], and several spontaneous ZS case reports have mentioned the persistence of symptoms for weeks until an effective acaricide treatment was applied [8,9,10]. These reports suggest that scabies mites’ needs may be fulfilled in human skin, too. Transmission from humans to animals has also been documented in mountain gorillas [11]. Furthermore, *S. scabiei* cross-transmission between different animal species has been reported in more than 50 species [12,13], under natural or human-driven conditions, highlighting the pronounced epidemiological plasticity of this ectoparasite. 

Online bibliographic enquiries using terms such as “pseudoscabies” or “ZS” point out that literature on ZS is not as abundant as expected and is partially outdated and difficult to retrieve. In addition, reviews on ZS episodes are lacking, except for sporadic contributions focusing on zoonotic canine scabies [10,14,15]. Because of all this, flaws exist in multiple aspects of ZS global comprehension, ranging from the proper use of terms (e.g., pseudoscabies; see below) to the epidemiology (e.g., diversity and relative importance of animal sources, or mite “strains” involved), the diagnosis and the treatment to recommend. This review is conceived as a narrative collection and critical analysis of the available literature on *S. scabiei* zoonotic episodes, focusing on the source of the outbreaks and the circumstances leading to the transmission of the parasite.

## 2. Methodology

We searched and selected relevant papers through three electronic databases via Web and interlibrary services (Scopus, Web of Science and Google Scholar) published until November 2021 with no time or language limits. The search strategy included different combinations of two or more of the following key words: “*Sarcoptes scabiei*”, “scabies”, “human”, “animals”, “sarcoptic mange”, “pseudoscabies”, “zoonosis”, “zoonotic disease”. Initially, titles and abstracts were screened, identifying articles for their relevance to the present topic. Finally, 46 relevant full papers and one conference abstract were analyzed. Complementary information was obtained from the online version of a monograph by Delafond and Bourguignon from the late 1800s [7].

## 3. Overview of Zoonotic Scabies Episodes

On a global scale, ZS episodes are associated with five continents (Africa, North and South America, Europe and Asia) (Figure 1), with the majority of reports originating from North America and Asia (Table 1).

According to the data shown in Figure 2, both domestic and wild animals are involved in the transmission pathways (Figure 2). The temporal range of collected articles is seventy years (1949–2020) with no clustering pattern along years, highlighting a low, though stable global reporting rate of ZS. As opposed to human-derived scabies, ZS seems to follow a scattered pattern of spreading associated with high-risk circumstances, rather than a cyclical pattern.

Pet owners are an important ZS target. So far, five pet species, namely dogs, cats, miniature pigs, horses and rabbits have been identified as sources for their owners(Figure 2). Dogs are by far the most reported source of ZS episodes worldwide (Figure 2). It is not clear whether this is a bias due to the fact that dogs are the most popular pet in several countries or because direct contact is particularly frequent and intimate in the case of dog ownership, or a combination of both. As a matter of fact, while sarcoptic mange is commonly diagnosed in stray, low-care or sheltered dogs [60], the condition is relatively rare amongst cats, another popular pet worldwide [59]. However, compared with other carnivore hosts of *S. scabiei* (e.g., red foxes, *Vulpes vulpes*), dogs are less prone to develop the crusted “Norwegian” form of the disease, which is associated with high mite density and a subsequent greater infectious potential than other sarcoptic mange presentations [61]. Worthy of note, in five studies, the canine origin of the ZS outbreak was associated with the recent adoption of a puppy with crusty lesions and/or harboring abundant mites [9,10,17,18,20]. The authentic zoonotic nature remains debatable in the case of a 14-year-old girl with an eight-year history of disseminated pruritic crusted lesions in a household where scabietic dogs were also present [27].

ZS originating from livestock is also a well-known condition, earning appellations such as “goat handler’s itch”, “dairyman’s itch”, “buffaloman’s itch”, “pig handler’s itch” and “cavalryman’s itch” [5,38,39,55]. Reports involve different livestock species, namely goats, sheep, pigs, cows, alpacas, llamas and water buffaloes (Figure 2), [37,38,39,48,49,55]. These ZS cases mainly include operators who were in intimate contact with infested livestock because of their daily work (e.g., milkmen, breeders and animal attendants). On a global scale, it is reasonable to assume that ZS cases linked with livestock animals are underreported compared to ZS cases related to pets. Nonetheless, compared with literature analyzed in this review, the rich and varied caseload in Delafond and Bourguignon [7] suggests that ZS of livestock origin is now uncommon in the Western world compared to the 19th century, likely due to the different attitude and management of animals and the availability of efficient and user-friendly acaricides. This is particularly true for ZS of horse origin, which collapsed in Europe in parallel with the contraction of the use of horses as military and non-military work animals. Sarcoptic mange has turned into a rare condition in equines throughout developed countries, and recent cases reported in Europe have instead been traced back to spillover contact with mangy foxes [34,45,46,62].

Wildlife does not appear to be a common source of ZS, as only nine species in different parts of the globe have been found to be responsible for mite transmission to humans (Figure 2), although this might be another bias due to the infrequent skin-to-skin contacts between humans and wild animals. Red foxes were associated with human scabies in five cases [8,33,34,35,36], in both urban and rural contexts. Worthy of note is an outbreak involving four people on a farm where a moribund fox with generalized lesions had sought shelter [34]. In this episode, people became infested through direct contact with different species of domestic animals (pigs, goats, dogs, horses, oxen) that, in turn, had previous contact with the scabietic fox. However, in other wildlife-derived ZS cases, the affected individuals were involved in wildlife-related occupations as keepers, veterinarians or specialized operators, or were private citizens who found themselves rescuing a scabietic animal or handling a fresh carcass [8,36,43,44,47,53,63]. Reportedly, wearing gloves did not guarantee protection in all cases [44,47], possibly due to the enormous number of mites crawling on the skin surface of source individuals affected by generalized crusted scabies. Beyond the literature, the authors have personal experience that ZS is not uncommon among hunters and gamekeepers in areas of Central and Southern Europe where mountain-dwelling ruminants are endemically infected with sarcoptic mange [44,47]. However, to the authors’ knowledge, no ZS cases have been reported in wild boar hunters, although the disease is observed in wild boar throughout Europe [13,64,65]. Delafond and Bourguignon [7] mention a single ZS case following contact with the skin of a scabietic wild boar in Germany. In the same monograph, a ZS outbreak was reported in zookeepers after contact with affected lions (*Panthera leo*), a hyena (*Hyaena hyaena*) and a bear (*Ursus arctos*) within weeks of importation to France.

On occasion, people were infected by other people who had acquired scabies from a wild or domestic animal source [19,43,44,47], implying the possibility of a person-to-person transmission of animal-derived mites. Symptoms in these people were apparently milder than in source contacts and resolved without any treatment within two weeks of exposure.

ZS episodes in this review either involved single patients (*n* = 21/47 citations) or occurred in the form of small outbreaks, with clinical signs appearing in more members of the same family (*n* = 20/47 citations) [9,25,30] or in professionals who handled the same infested animal(s) in the same working environment [40,43,55] (Table 1).

## 4. Characterization of *S. scabiei* Mites in Zoonotic Scabies Episodes

It is accepted that the study of morphology is of limited, if any, value in characterizing the different host- or host-group-adapted variants (“*varietates*”) recognized so far within the globally distributed *S. scabiei*. Consequently, the mite variants involved in ZS episodes have generally been identified on the basis of patient history and epidemiological background. While this is still a primary approach, sharing of similar mite variants can now be demonstrated on the basis of robust molecular evidence. Amongst several genetic markers available, microsatellites were shown to be the most informative and accurate for the purpose of characterizing interspecific cross-transmission of *Sarcoptes* mites [61,66,67,68]. What is promising is that last-generation techniques, such as mitochondrial metagenome sequencing [69] or whole mite genome sequencing [70], can now be used to further clarify the genetic diversity within mite species.

Walton et al. [71] were the first to show that randomly sampled Australian aborigine patients were occasionally infested by *Sarcoptes* mites genetically clustering with those collected from sympatric community dogs. More recently, in the context of the aforementioned ZS episode that occurred on a farm in Switzerland [34], mites collected from different and zoologically distant domestic hosts were found to be genetically similar to those isolated from the scabietic fox that likely prompted the outbreak. In Italy, Moroni et al. [35] provided unambiguous evidence that two patients and the scabietic fox they incautiously rescued, harbored the same vulpine variant of the pathogen. The authors are not aware of any studies based on other genetic markers (namely mitochondrial and ribosomal), in which mites collected on the occasion of spontaneous ZS episodes have been characterized. However, the study of polymorphisms in the cytochrome c oxidase subunit 1 gene [72] identified remarkable differences between sympatric and allopatric mites of human origin and showed that some human-derived isolates clustered with mites of animal origin, e.g., dog, pig and raccoon dog (*Nyctereutes procyonoides*), suggesting that *Sarcoptes* is anything but a panmictic taxon and that *Sarcoptes* taxonomy is not as simplified as the “traditional” denomination that the clear host-based variants would suggest.

Despite recent genetic advances, *Sarcoptes* molecular typing remains challenging for several reasons, including the difficulty of isolating individual mites from skin scrapings or biopsies and the low success rate of DNA extraction and PCR amplification of mite DNA [67,73].

## 5. Diagnosis of Zoonotic Scabies

ZS diagnosis is often based on compatible signs coupled with consistent historical features, as the visualization of animal-derived mites on a patient can be challenging but it is still described in the literature [10,15,26,43].

ZS usually manifests as an intensely pruritic papulovesicular rash, affecting areas such as the trunk, abdomen, forearms, thighs and legs, while sparing those parts of the body (e.g., palms of hands, soles of feet and genitalia) that are frequently involved in human scabies by *S. scabiei* var. *hominis* [15,74,75] (Figure 3). However, some exceptions are known in children exposed to scabietic pets, where the distribution pattern of skin lesions, involving the palms, web of fingers, head and neck, may resemble human scabies [6,26,28]. Body areas in contact with the animal source are the first affected, but other areas may be involved later [43,47]. The different distribution of skin lesions between human and ZS should be further investigated through systematic clinical and dermatological examinations to draw a valid conclusion on the different immuno-pathological processes behind it. In a more recent paper [10] in which the authors used dermoscopy, emphasis was put on the diagnostic significance of papules showing a curvilinear crust over a yellow background, possibly corresponding to the remnant of a short superficial burrow. Typically, the onset of itching is much more rapid (e.g., hours or a few days) than in the case of human scabies (i.e., weeks), and a nocturnal exacerbation is often reported [19,43,47,49].

In one of the experimental infections mentioned previously [6,7], the prototypical lesions were erythematous papules and vesicles and burrows. Pruritus was evident within 24 h in both volunteers, and histologic biopsy findings were similar to those observed in human scabies.

As is well known, the search for the parasite on patients may be unsuccessful and the final diagnosis is usually based on clinical signs such as typical pruritic skin lesions (19/47 cases investigated in this review; see Table 1) and/or on the success of clinical response after application of an acaricide treatment (ex-juvantibus). Nonetheless, the presence of burrows with mites and/or eggs has been reported in ZS patients following skin scrapings or dermoscopic examination [9,26,28,35,53]. It was shown in an experiment [6] that more than half of the adult female mites deposited on the skin of two volunteers managed to survive until the end of the trial (96 h), and some laid viable eggs.

In two case reports, *S. scabiei* was erroneously diagnosed by the authors as the primary cause of a pruritic rash in a 56-year-old man and in two farmers, respectively [76,77]. However, pictures in the articles showed a chigger mite [76] and a Chorioptes spp. mite [77], suggesting that ZS misdiagnosis may occur even when mites are collected and observed under a microscope, if for some reason physicians are not familiar with zoonotic ectoparasites.

Considering the self-limiting nature of the majority of ZS episodes, more attention should be drawn to prevention and early diagnosis. This is particularly important in vulnerable populations such as the elderly, the young and immunocompromised patients, in whom early diagnosis would make a difference. Pruritus conditions can be easily misdiagnosed in these categories as psychogenic, degenerative or senile conditions, rather than of parasitic origin. It is therefore essential to include scabies in the differential diagnosis of any persistent non-classic pruritus erythematous rash, especially in settings where the patient may have had contact with potentially mangy animals.

## 6. Treatment and Control

Because of the origin of ZS and its usually self-limiting nature, resolution of symptoms does not necessarily imply the etiological treatment of the patient, as evidence exists that treating and/or interrupting direct contact with the animal source(s) are sufficient for this purpose. Nonetheless, some dermatologists testify that patients are rarely willing to wait for the disease to run its course [15].

Regarding the benefits of applying an etiological treatment, Smith and Claypoole [28] reported a shorter resolution time in ZS patients receiving a single 24 h topical lindane treatment compared with untreated patients, but this was not tested in a randomized trial scheme. A relatively long resolution time (two weeks) was instead observed in an adult patient treated with 10% lindane lotion [44], and also failure to clear the rash was also reported after massive exposure to an infested fox [8]. Application of 5% permethrin cream represents another recommended treatment option [9,10,45], but the resolution times are unfortunately not available, with the exception of a single report that mentions clinical resolution in approximately one week after treatment application, which is less than the traditionally reported 2–3 weeks characterizing the spontaneous course of the disease [43]. From all the above, it is clear that additional evidence-based information on the efficacy of current etiological treatment protocols is necessary. Treatment of symptoms involving the administration of sedative antihistamines and/or emollient cream, can help to relieve itching, reduce scratching and preserve sleep time.

The identification and successful etiological treatment of animal source(s) is crucial not only to establish a correct diagnosis but also to avoid potentially unsafe over-treatment in patients with recurrent symptoms. Effective treatment protocols against sarcoptic mange are available in several domestic and wildlife species, including the new drug class of isoxazolines (e.g., fluranaler, afoxolaner) [78,79,80], although the treatment may be challenging in animal patients with generalized crusted mange [49,81] and in susceptible free-ranging wildlife [82,83].

## 7. Knowledge Gaps and Conclusions

“Pseudoscabies” is a broad definition describing a “skin eruption caused by mites for which humans are not the normal host” [4,84]. Since it can be applied to ZS as well as to several other dermatological conditions that clinically resemble human scabies but have a different etiology (e.g., those attributable to bird or rodent mites, namely *Dermanyssus gallinae* and *Ornithonyssus sylviarum* [84,85], or to environmental and plant mites, namely *Trombicula* spp., *Pyemotes ventricosus* [4]), we believe that such a definition is ambiguous and adds more confusion to the already complex scenario of scabies sources and epidemiology. We therefore encourage dermatologists and parasitologists to prioritize the term “zoonotic scabies”, specifying the animal source of the infection, if available.

Descriptive data on ZS episodes are still limited. The number of events may be greatly underestimated considering that the scientific literature in this review also includes outdated case reports, anecdotal reports and grey literature. Moreover, most of the cases were only reported as a supplement or added details in the frame of another study, and not as a primary description of ZS, which explains why keywords such as “pseudoscabies” or “zoonotic” did not appear in the abstract. This might also be attributed to the fact that ZS originating from numerous sources has already been described, and new reports in the scientific literature would now seem repetitive or obsolete when known animal sources are involved. Yet, ZS remains a neglected zoonosis, with no surveillance nor coordinated reporting system on a regional or global scale. Thus, information on its prevalence and spread can only be extrapolated from the available scientific literature or from anecdotal reports, with implicit inaccuracy.

With the advent of next-generation sequencing such as whole-genome sequencing, new methodologies will be available to explore genomic differences among animal and human *S. scabiei* lineages in detail [70]. To enhance such new opportunities, collaboration between scabies researchers all over the world would be essential in order to share samples from different animal species and geographical origins and to standardize techniques and data interpretation. More first-hand information by dermatologists is also warranted to understand the variability in the morbidity, disease severity and timing of the response to treatment among people infected with different animal-derived strains.

Finally, we believe that key priorities in understanding ZS are represented by an increased surveillance in high-risk occupations and, on the occasion of outbreaks, improved communication between animal and human health specialists, in line with the One Health approach.

## Figures and Tables

**Figure 1 pathogens-11-00213-f001:**
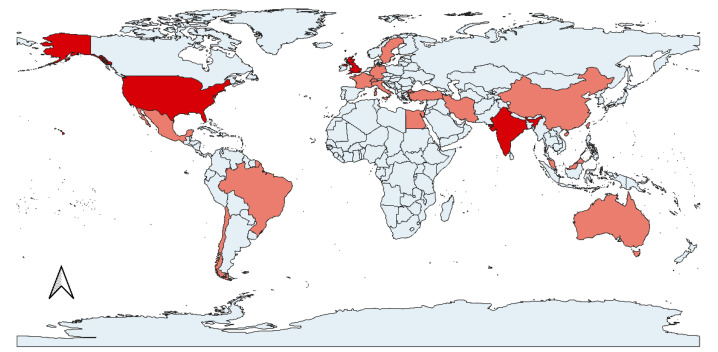
World map showing countries with cases of zoonotic scabies reported in the scientific literature (dark red: 4 or more articles associated with this country; light red: fewer than 4 articles).

**Figure 2 pathogens-11-00213-f002:**
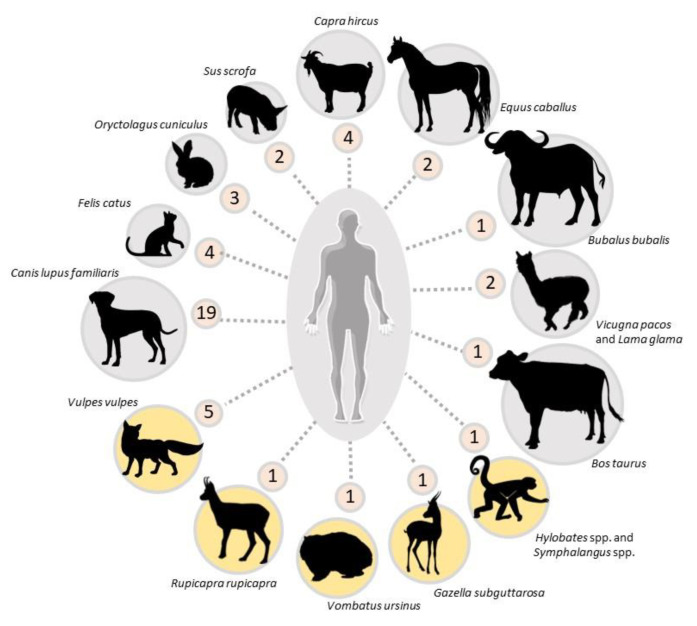
Animal species origin of zoonotic episodes of scabies represented by number of articles/animal species collected in this review. Wildlife species are highlighted in yellow. The ZS episodes described by Delafond and Bourguignon [7] were not included in this figure but are mentioned in the main text.

**Figure 3 pathogens-11-00213-f003:**
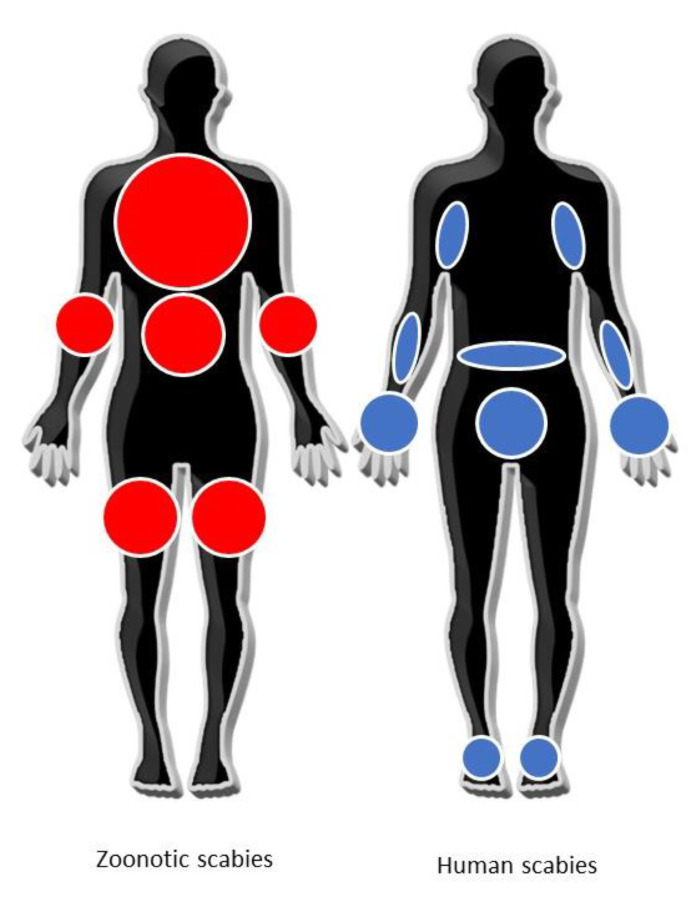
Distribution of typical clinical skin manifestation in zoonotic (in red) and human (in blue) scabies, adapted from Engelman et al. [75].

**Table 1 pathogens-11-00213-t001:** Reported episodes of zoonotic scabies. Relevant information on the source of the transmission, country of the episode, epidemic aspect (when more than 2 people were infected) and type of diagnosis. ZS cases observed by Delafond and Bourguignon [7] are not included in this table.

	Country	Epidemic Aspect (Number of People Infected)	Diagnosisin Humans	References
**Contact with dogs (*Canis lupus familiaris*)**				
	Turkey	no	Dermoscopy	[10]
	USA	yes (*n* = 9)	Clinical	[16]
	USA	yes (*n* = 15)	Skin scraping	[17]
	South Korea	no	Skin scraping	[18]
	USA	yes (*n* = 10)	Clinical	[19]
	Chile	yes (*n* = 7)	Skin scraping	[9]
	South Korea	No	Skin scraping	[20]
	UK	No	Clinical	[21]
	Brazil	yes (*n* = 58 *)	Skin scraping/clinical	[22]
	USA	-	-	[23]
	Egypt	-	-	[24]
	USA	yes (*n* = 4)	Skin scraping	[25]
	USA	-	Skin scraping	[26]
	Mexico	No	Skin scraping	[27]
	USA	yes (*n* = 22 *)	Clinical/ex- juvantibus	[28]
	India	No	-	[29]
	USA	yes (*n* = 7)	Skin scraping	[30]
	UK	No	Clinical	[31]
	USA	yes (*n* = 67 *)	-	[32]
**Contact with red foxes (*Vulpes vulpes*)**				
	Germany	No	Clinical/ex-juvantibus	[33]
	Switzerland	yes (*n* = 4)	Dermoscopy	[34]
	USA	No	Clinical/ex-juvantibus	[8]
	Italy	No	Dermoscopy	[35]
	Sweden	-	-	[36]
**Contact with Bovidae**				
Cow (*Bos taurus*)	-	No	Skin scraping	[37]
Water buffalo (*Bubalus bubalis*)	India	yes (*n* = 35)	Skin scraping and clinical	[38]
Goat (*Capra hircus*)	India	yes	-	[39]
	India	yes (*n* = 13)	Skin scraping	[40]
	India	No	Skin scraping	[41]
	India	yes	-	[42]
Chamois (*Rupicapra rupicapra*)	Italy	yes (*n* = 7)	Clinical	[43]
Goitrerd gazelle (*Gazella subgutturosa)*	Iran	yes (*n* = 6)	Skin scraping	[44]
**Contact with horses (*Equus caballus*)**				
	UK	No	Clinical	[45]
	UK	No	Clinical	[46]
**Contact with wombats (*Vombatus ursinus*)**				
	Australia	yes (*n* = 3)	Clinical	[47]
Contact with Camelidae				
Alpaca (*Vicugna pacos*)	UK	No	Clinical/ex-juvantibus	[48]
Llama, alpaca (*Lama glama, Vicugna pacos*)	Germany	No	Clinical	[49]
**Contact with rabbits (*Oryctolagus cuniculus*)**				
	South Korea	No	Ex-juvantibus	[50]
	China	-	-	[51]
	China	-	-	[52]
**Contact with primates**				
Gibbon (*Hylobates leuciscos*)	UK	yes (*n* = 10)	Dermoscopy	[53]
**Contact with pigs (*Sus scrofa*)**				
	Switzerland	No	Clinical	[54]
	India	yes (*n* = 30)	Skin scraping/clinical	[55]
**Contact with cats (*Felis catus*)**				
	UK	No	Clinical	[56]
	Indonesia	No	Clinical	[57]
	Taiwan	yes (*n* = 5)	Clinical	[58]
	Australia	No	Clinical	[59]

* more than a single epidemic event.

## Data Availability

The datasets used and analyzed in the present study are included in this article.

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
