# Peer review of "Zoonotic Episodes of Scabies: A Global Overview"

_pathogens, 2022, doi:10.3390/pathogens11020213_

Round 1
Reviewer 1 Report
Pathogens 1558173 peer review
Brief summary
The aim of this review was to collect the existing literature on the zoonotic transmission of scabies to humans globally and provide a critical review covering the source of the outbreak, the circumstances leading to the transmission of the parasites, diagnosis, identification of the Sarcoptes scabiei strain involved and the treatments applied.
This article is a narrative global review of zoonotic scabies. The review selected papers through Scopus, Web of Science and Google Scholar using appropriate search terms and the papers were screened for appropriateness. I presents a world map showing the geographical distribution of zoonotic scabies, a diagram showing the species of animal sources and the number of episodes of each in the literature, a summary of the literature on which this was based, and the difference in the distribution of scabies mites on the human host between zoonotic scabies and human scabies.
The review highlights the scarcity of publications on zoonotic scabies. In particular it is very unclear which strains occur on human and non-human hosts in different regions of the globe. The review highlights the opportunity to develop standardised nucleic acid techniques to address this question.
The manuscript is clear, relevant for the field and presented in a well-structured manner. The figures and tables are useful and appropriate. A big gap in knowledge is identified. The conclusions are supported by the literature.
To my knowledge, there are no recent reviews in the literature on this topic with a similar scope.
The cited references cover a wide time period. This is appropriate given the neglect in this area, and the need to collect as much relevant information as possible.
Rating the manuscript:
- Originality/Novelty: Yes, the question is original and well defined. The results provide systhesis of current knowledge?
- Significance: Are the results interpreted appropriately? Are they significant? Are all conclusions justified and supported by the results? Are hypotheses and speculations carefully identified as such? Yes
- Quality of Presentation: Is the article written in an appropriate way? Are the data and analyses presented appropriately? Are the highest standards for presentation of the results used? Yes
- Scientific Soundness: Yes. The authors are systhesising existing knowledge.
- Interest to the Readers: The conclusions are interesting for the readership of the Journal, and should attract a wide readership as scabies is a global infectious disease and is very infectious, and zoonotic scabies may be more a important source of scabies than our current knowledge suggests.
- Overall Merit: Yes, there an overall benefit to publishing this work? Does the work provide an advance towards the current knowledge. Yes. The authors have addressed an important long-standing question by gathering together existing scant published information.
- English Level: Yes the English language used is appropriate and understandable.
Ethics
This is not applicable as the manuscript is based on already published material.
Interest
This article is of interest to a large number of people because the zoonotic scabies is poorly understood and zoonotic scabies infections are often considered trivial.
Specific comments
Line 71: “monography” should be “monograph”. Please change everywhere where it occurs in the manuscript.
Lines 74 and 74: six continents: Oceania isn’t a continent. If you call them six world regions, then the list holds true.
Line 77: add “Table 1” after “Figure 1”.
Recommendation
This is an important work and should be published with very minor changes as listed under specific comments above.
Author Response
Thank you for the positive comments. We appreciate the acknowledgement of the work done.
Line 71: “monography” should be “monograph”. Please change everywhere where it occurs
in the manuscript.
Response. Modified as suggested
Lines 74 and 74: six continents: Oceania isn’t a continent. If you call them six world
regions, then the list holds true.
Response. We amended the term as “five continents”.
Line 77: add “Table 1” after “Figure 1”.
Response. We changed the Table position as suggested.
Reviewer 2 Report
Review
on article of B. Moroni, L. Rossi, C. Bernigaud and J. Guillot
“Zoonotic episodes of scabies: a global overview”
The manuscript is presented a well-illustrated overview of zoonotic cases of scabies reported in the world scientific literature. The authors have collected all the available scientific literature on this theme. Congratulations to the authors. The article is of undoubted interest.
However, I believe that the paper needs to be improved in order to be published. My comments and suggestions are as follows:
- The main remark is that the article needs the traditional sections Methods and Results. In Methods, you should arrange search strings (remake and expand existing ones in Introduction). (For example, see recently published in Pathogens article: Patel A., Jenkins M., Rhoden K., Barnes A.N. 2022. A Systematic Review of Zoonotic Enteric Parasites Carried by Flies, Cockroaches, and Dung Beetles. Pathogens. 11, 90. https://doi.org/10.3390/pathogens11010090)
- Lines 2, 21, 84, 133, 144 and further in the manuscript text – It is better to use “case or cases” rather than “episode or episodes”.
- In the Abstract (Line 15) should write the full Latin name of the mite for the first time.
- At the first mention of an animal species (mites and its vertebrate hosts) in the text, its full Latin name with the author and year of description should be given. Further, at the second mention of the Latin name of the parasite, the generic name of the parasite is reduced to one letter, the author and the year are not given. When first mentioned Latin mite name in Introduction should be indicated the systematic taxon in brackets – (Sarcoptiformes: Sarcoptidae).
- Table 1 should appear immediately after the first mention to it (Line 76).
- Lines 159-163 – The sentences should be rephrased. Unsuccessfully used terms for ticks are “variants” (“varieties”), “mite strains”, “mite lineages”. It may be better to use the terms “subspecies”, “intraspecific groups” or something else?
7. Line 263 – wrong section number
- Line 301 – probably better to write “when more than two people were infected”?
In conclusion, I express my opinion – the manuscript can be published, but minor corrections are needed.
Author Response
Thank you very much for your positive comments and suggestions which have been helpful and constructive to revise and improve the paper. The manuscript has been modified according to your comments and the revised text is in track change.
The main corrections and the detailed responses to the reviewers’ comments are provided below. We hope that you will be satisfied with the modifications and we look forward to hearing from you.
The authors
The main remark is that the article needs the traditional sections Methods and Results. In Methods, you should arrange search strings (remake and expand existing ones in Introduction). (For example, see recently published in Pathogens article: Patel A., Jenkins M., Rhoden K., Barnes A.N. 2022. A Systematic Review of Zoonotic Enteric Parasites Carried by Flies, Cockroaches, and Dung Beetles. Pathogens. 11, 90. https://doi.org/10.3390/pathogens11010090)
Response. This was one of the main points that the authors discussed before the submission of the manuscript. We added a “methodology” chapter (number 2). However, given the format of our review (which is not a systematic one), we decided not to add a “results” section, as we present observations and conclusions in the following chapters. The format that we chose is accepted by the journal guidelines. https://www.mdpi.com/journal/pathogens/instructions#back
This choice is based on the fact that most of the articles that are included in the review were not retrieved by an automated “search string collection” on electronic databases, but selecting paper by paper and revising all the bibliography that could lead to the targeted topic.
This is probably due to the nature itself of the topic, which was in many papers only mentioned within the discussion, and not presented as a result. Unfortunately, this aspect did not allow to follow a reproducible methodology, which is the main requirement for systematic reviews.
We discuss about this point in the conclusion (lines 290-298):
“Descriptive data on ZS episodes are still limited. The number of events may be greatly underestimated considering that the scientific literature in this review also embraces outdated case reports, anecdotal reports, and grey literature. Moreover, most of the cases were only reported as supplement or added details in the frame of another study, and not as primary description of ZS, which explains why keywords such as “pseudoscabies” or “zoonotic” did not appear in the abstract. This might be also attributed to the fact that ZS originating from numerous sources has already been described, and new reports in the scientific literature would now seem repetitive or obsolete when known animal sources are involved.”
Lines 2, 21, 84, 133, 144 and further in the manuscript text – It is better to use “case or cases” rather than “episode or episodes”.
Response. We modified the text accordingly, although in the title we would like to keep the form “Zoonotic episodes of scabies” as we refer to the general occurrence of the disease and not to a specific case.
In the Abstract (Line 15) should write the full Latin name of the mite for the first time.
Response. We modified the text accordingly.
At the first mention of an animal species (mites and its vertebrate hosts) in the text, its full Latin name with the author and year of description should be given. Further, at the second mention of the Latin name of the parasite, the generic name of the parasite is reduced to one letter, the author and the year are not given. When first mentioned Latin mite name in Introduction should be indicated the systematic taxon in brackets – (Sarcoptiformes: Sarcoptidae).
Response. We added the systematic taxon of Sarcoptes scabiei in brackets + the author and year of description as suggested, in line 32. We also added Latin names for all the vertebrate hosts included in the manuscript. Regarding the author and the year of description of each animal species mentioned in the text, we believe that this is not necessary as the journal guidelines do not require it.
Table 1 should appear immediately after the first mention to it (Line 76).
Response. We modified the table position accordingly.
Lines 159-163 – The sentences should be rephrased. Unsuccessfully used terms for ticks are “variants” (“varieties”), “mite strains”, “mite lineages”. It may be better to use the terms “subspecies”, “intraspecific groups” or something else?
Response. We agree with the reviewer that the terms used might sound confusing.
In fact, this terminology issue represents an ongoing debate among Sarcoptes scientists.
Traditionally, the term “variant” is used to specify host-specific Sarcoptes scabiei (Arlian et al., 1989), although we might say that this subdivision is outdated and does not completely or always fit with the parasite epidemiology, but it is still accepted and widely used in the scientific literature.
In the revised version of the review, we decided to use the single term “variant”.
Line 263 – wrong section number
Response. We corrected the number.
Line 301 – probably better to write “when more than two people were infected”?
Response. We modified the text as suggested.
Reviewer 3 Report
This manuscript addresses a mini review about zoonotic episodes of scabies. A bibliographic revisión, with different key words, was carried out searching for Zoonosis Scabies episodes.
I think this paper includes data of publication quality, and the types of analyses are properly used to evaluate the results. The revision shows an exhaustive update. The Figures and Tables are necessary and the references adequate. Furthermore, clinical practitioner must read this paper because some of them think that is not possible transmission from animal hosts to humans. Even, I suffered an infection from dog pet but doctor did not admit this way of transmission.
Nevertheless, molecular studies are necessary to corroborate this point.
My experience working on so tiny mites makes me agree with the author about the difficulty of handling, DNA extraction, lack of molecular data and therefore to design efficient primers from species of Sarcoptes scabiei from different hosts or hosts from differents geographic origin. That is why I think this is an interesting paper for acarologists working on this kind of mites. On the other hand, if the use of this molecular techniques is not established as general, the problem of lack of molecular data will go on.
Therefore, I can conclude that this ms is acceptable in its present version.
Author Response
Thank you very much for your positive feedback.
We agree with you that one of the main problem on Sarcoptes scabiei epidemiology remains the genetic identification of different variants/lineages.
We auspicate further collaborations between clinicians and parasitologists to overcome the knowledge gap that are currently present in the scientific literature.
Reviewer 4 Report
Scabies is a common parasitic skin condition that causes considerable morbidity globally. The short literature review focuses on zooanthroponotic transmission of S. scabei and does not deal with dermatophytosis of animal origin. The review taste like a systematic review of the literature although in its form it does not fulfill the criteria for systematic review and that is a pity. A pity because this review is scientifically sound and provide a new focus on this disease. Lastly in my opinion this sort of subject is more appropriate for a scientific journal like International Journal of Environmental Research and Public Health than for Pathogens. Firstly, because molecular, cellular, and immunological determinants of Pathogenicity are not under the scope of the review. Secondly, because most of the review deal with the epidemiological life cycle of S. scabei. Nevertheless, the review retained its interest and deserves publication.
Remarks
Chapter 3 “Characterization of S. scabei….” This part of the review is crucial for the reader, and information that helps them to get a clearer view of the subject is of importance. This chapter begins with information on the difficulties to address S. scabei morphological identification. I think that pictures of mites will be of interest to the reader especially if they highlight such difficulties to differentiate populations of S. scabei from animal and human origin.
In the same chapter, the discussion on molecular evidence of population clustering between mites of animal (wild life-pet) and human ones, deserves a discussion on criteria that have governed the delineation of the lineages/var.
Is there some molecular marker and primer developed to diagnose Human scabies, if yes do these primers or markers work for zoonotic scabies?
Is there some immunological marker available?
Do mites of animal origin that can undergo a complete development life cycle in human subjects, show differences in their development laps (egg to adult)? I mean in terms of times of transition. In this chapter, a figure showing this development life cycle would be of interest to readers
Chapter 4. Does the literature survey performed in this review support the distribution of typical clinical skin manifestation in zoonotic and human scabies, proposed by Engelman. What would be the determinant that explain such differences in the body distribution of mites?
Chapter 5. Conclusion. For me, but I am not an S. scabei specialist, the most important challenge would be to investigate the population genetics of S. scabei in animals and humans, and to re-address the taxonomy of this arthropod. These would clearly help to prompt the development of new and more efficient diagnostic tools and help to address the risk of scabies infection.
Author Response
Thank you very much for your positive comments and suggestions which have been helpful and constructive to revise and improve the paper. The manuscript has been modified according to your comments and the revised text is in track change.
The main corrections and the detailed responses to the reviewers’ comments are provided below. We hope that you will be satisfied with the modifications and we look forward to hearing from you.
The authors
Scabies is a common parasitic skin condition that causes considerable morbidity globally. The short literature review focuses on zooanthroponotic transmission of S. scabei and does not deal with dermatophytosis of animal origin. The review taste like a systematic review of the literature although in its form it does not fulfill the criteria for systematic review and that is a pity. A pity because this review is scientifically sound and provide a new focus on this disease. Lastly in my opinion this sort of subject is more appropriate for a scientific journal like International Journal of Environmental Research and Public Health than for Pathogens. Firstly, because molecular, cellular, and immunological determinants of Pathogenicity are not under the scope of the review. Secondly, because most of the review deal with the epidemiological life cycle of S. scabei. Nevertheless, the review retained its interest and deserves publication.
Response. Thank you for the positive feedback! We agree with the comment on the systematic “taste” of this work. As replied to Reviewer 1, it was not possible to achieve a systematic methodology on this topic, as the majority of the papers collected did not contain information on pseudoscabies in the abstract/title, but they only mentioned the cross-infection occurrence within the paper.
Chapter 3 “Characterization of S. scabei….” This part of the review is crucial for the reader, and information that helps them to get a clearer view of the subject is of importance. This chapter begins with information on the difficulties to address S. scabei morphological identification. I think that pictures of mites will be of interest to the reader especially if they highlight such difficulties to differentiate populations of S. scabei from animal and human origin.
Response. As we mentioned in the manuscript: “the study of morphology is of limited, if any, value in characterizing the different host- or host-group-adapted variants”. Indeed, morphology of S. scabiei does not support any kind of substantial differences among distinct host-specific strain, and this is why we think that a picture of S. scabiei would not improve the understanding of the manuscript.
In the same chapter, the discussion on molecular evidence of population clustering between mites of animal (wild life-pet) and human ones, deserves a discussion on criteria that have governed the delineation of the lineages/var. Is there some molecular marker and primer developed to diagnose Human scabies, if yes do these primers or markers work for zoonotic scabies?
Response. To the authors’ knowledge, there are no specific markers that would work specifically (and exclusively) for human S. scabiei. In fact, the same molecular markers used for animal-derived S. scabiei work exactly the same for human-derived Sarcoptes. Moreover, such molecular markers are usually developed and used for population genetics analysis, and not for the routine diagnosis of scabies. The authors have been working on microsatellite markers for S. scabiei, and in one of the cited abstract (n. of reference: 35), we showed molecular typing of Sarcoptes mites from a fox and two women that were infected by the fox using a panel of 10 microsatellite markers. The results of the study highlighted that the molecular profile of fox-derived mites were consistent with those isolated from the women.
Is there some immunological marker available?
Response. There are several recent studies on the immune response to Sarcoptes scabiei in humans (Bhat et al. Parasites & Vectors (2017) 10:385 DOI 10.1186/s13071-017-2320-4), but we didn’t find any specific immunological marker available.
Do mites of animal origin that can undergo a complete development life cycle in human subjects, show differences in their development laps (egg to adult)? I mean in terms of times of transition. In this chapter, a figure showing this development life cycle would be of interest to readers
Response. Thank you for pointing out this interesting aspect of zoonotic scabies. There are indeed very few studies investigating the different development timing of human and zoonotic scabies. In order to perform such study in controlled conditions in humans, an experimental infection would be needed, which is nowadays quite a challenge (not to say impossible!) from the ethical point of view. In the study by Estes et al., (1983), an experimental infection of canine scabies in a man has been performed. Estes et al. stated that “This experiment demonstrates that canine S. scabiei were able to survive in human skin for at least 96 hours in the undisturbed state that existed underneath the disc. The mineral oil preparations demonstrate that the mites burrowed, defecated, laid up to nine eggs which developed normally, and that some eggs subsequently hatched. This suggests that the mite nutritional needs were fulfilled in human skin. Whether a complete life cycle can occur on humans has yet to be determined.” To our knowledge, all other experimental studies performed on zoonotic scabies date back to the 19th century, and we did not find detailed information on zoonotic Sarcoptes life cycle but in Estes et al’s study, which unfortunately it is not sufficient to elaborate a figure.
Chapter 4. Does the literature survey performed in this review support the distribution of typical clinical skin manifestation in zoonotic and human scabies, proposed by Engelman. What would be the determinant that explain such differences in the body distribution of mites?
Response. The distribution of clinical skin manifestation proposed by Engelman was specific for human scabies (and did not report anything on zoonotic scabies), so we cannot compare the studies in this sense. What we found interesting to compare was the different distribution between classic and zoonotic scabies (based on information collected in the articles in our review). Unfortunately, there are no clinical systematic studies that discussed the different distribution of skin lesions between the two forms, so we don’t have evidence-based explanations for this. We added the following sentence in line 213-216:
“The different distribution of skin lesions between human and ZS should be further investigated through systematic clinical and dermatological examinations to draw a valid conclusion on the different immuno-pathological processes behind”
Chapter 5. Conclusion. For me, but I am not an S. scabei specialist, the most important challenge would be to investigate the population genetics of S. scabei in animals and humans, and to re-address the taxonomy of this arthropod. These would clearly help to prompt the development of new and more efficient diagnostic tools and help to address the risk of scabies infection.
Response. We agree with you that the most important and interesting challenge would be to further investigate population genetics of both, animal and human- derived S. scabiei. This is indeed one of the main goals of our international scientific network on Sarcoptes. It has been quite a challenge for us to create a solid and continuous collaboration between clinicians and researcher on this topic, as scabies (especially in developed countries) is usually treated as soon as the diagnosis is made, and the mites are seldom collected and stored from humans.
Reviewer 5 Report
Review for PATHOGENS (MDPI)
Zoonotic episodes of scabies: a global overview
The study is really interesting and shows a fairly comprehensive picture of the current situation with respect to zoonotic scabies. We can also agree with the authors' recommendation ‘We, therefore, encourage dermatologists and parasitologists to prioritize the term ‘zoonotic scabies’, specifying the animal source of infection, if available’, which clarifies the terminology used in clinical practice…
It is really interesting how few scientifically documented cases the authors managed to collect. Especially in comparison with the general idea and the impression that pseudoscabies is quite common, especially in pet and domestic animal holders, or in hunters. It would be useful to comment on this aspect in even more detail - it would be ideal to compare the scientific information contained in scientific articles with information from clinical practice. It is well known that these two sources can completely diverge and provide diametrically opposed information.
Furthermore, given the focus of the article, it would be appropriate to comment at least briefly on whether, in addition to the transition of animal ecotypes to humans, there can also be a mutual transition of animal ecotypes (10.1111/tbed.14082; although only in a minority of the described cases can it be truly stated that this is a transfer of one ecotype from a typical host to another (untypical) animal host; moreover, it is not clear whether in animals infected with an inappropriate ecotype, develop only a short-lived episodic matter that disappears on its own, similar to human pseudoscabies). It would also be appropriate to mention possible co-infection with human and zoonotic scabies. I believe that the recommendations would make an already very interesting article even more comprehensive.
15 Sarcoptes scabiei
19 Sarcoptes
20-21 the number of episodes would be more informative than the number of articles...
46 strains vs. var. / variants / varietates vs. lineages ???? – This should be clarified. It is quite understandable that even the authors are not entirely clear on what term to use, and therefore use the term "strains" in quotation marks. However, it is worth considering whether it would not be more appropriate to use some other designation. The term strain would be appropriate for microorganisms (or asexually reproducing organisms) - typically, for example, for protozoa, bacteria, etc. However, in the case of mites, these are rather populations adapted to their hosts... (form / ecoform / ecotypes… ???)
51 Anthropozoonosis – the definition is very ambiguous and differs from different sources; the understanding of this term is very different in different subdisciplines, and therefore it is better to avoid this word altogether and only use a descriptive form... that transmission from humans to animals has also been documented...
Author Response
Thank you very much for your positive comments and suggestions which have been helpful and constructive to revise and improve the paper. The manuscript has been modified according to your comments and the revised text is in track change.
The main corrections and the detailed responses to the reviewers’ comments are provided below. We hope that you will be satisfied with the modifications and we look forward to hearing from you.
The authors
The study is really interesting and shows a fairly comprehensive picture of the current situation with respect to zoonotic scabies. We can also agree with the authors' recommendation ‘We, therefore, encourage dermatologists and parasitologists to prioritize the term ‘zoonotic scabies’, specifying the animal source of infection, if available’, which clarifies the terminology used in clinical practice…
It is really interesting how few scientifically documented cases the authors managed to collect. Especially in comparison with the general idea and the impression that pseudoscabies is quite common, especially in pet and domestic animal holders, or in hunters. It would be useful to comment on this aspect in even more detail - it would be ideal to compare the scientific information contained in scientific articles with information from clinical practice. It is well known that these two sources can completely diverge and provide diametrically opposed information.
Response. We totally agree with you. To underline the need for more scientific investigations on ZS, we included the following sentences in the conclusion (lines 290-298):
“Descriptive data on ZS episodes are still limited. The number of events may be greatly underestimated considering that the scientific literature in this review also embraces outdated case reports, anecdotal reports, and grey literature. Moreover, most of the cases were only reported as supplement or added details in the frame of another study, and not as primary description of ZS, which explains why keywords such as “pseudoscabies” or “zoonotic” did not appear in the abstract. This might be also attributed to the fact that ZS originating from numerous sources has already been described, and new reports in the scientific literature would now seem repetitive or obsolete when known animal sources are involved.”
Furthermore, given the focus of the article, it would be appropriate to comment at least briefly on whether, in addition to the transition of animal ecotypes to humans, there can also be a mutual transition of animal ecotypes (10.1111/tbed.14082; although only in a minority of the described cases can it be truly stated that this is a transfer of one ecotype from a typical host to another (untypical) animal host; moreover, it is not clear whether in animals infected with an inappropriate ecotype, develop only a short-lived episodic matter that disappears on its own, similar to human pseudoscabies). It would also be appropriate to mention possible co-infection with human and zoonotic scabies. I believe that the recommendations would make an already very interesting article even more comprehensive.
Response. Thank you for this interesting insight. We added the following sentence (lines 53-56): “Furthermore, S. scabiei cross-transmission between different animal species has been reported in more than 50 species [12,13], under natural or human-driven conditions, highlighting the pronounced epidemiological plasticity of this ectoparasite.”
In the cited work (Escobar et al., 2021), it seems that some species are more prone to be infected by other Sarcoptes strains (namely, animals belonging to the Artiodactyla Order), although it is not clear whether this species might have different (untypical) forms of scabies, rather than forms caused by same species infections.
Response. Regarding possible co-infection of human -zoonotic scabies, unfortunately we are not aware of such cases, although we do not exclude the possibility of this occurrence, which might have not been reported in the scientific literature yet.
15 Sarcoptes scabiei
Response. Modified as suggested
19 Sarcoptes
Response. Modified as suggested
20-21 the number of episodes would be more informative than the number of articles...
Response. We think that the number of articles is the most reliable information to give to the reader, as in some articles more than a single outbreak was described, and in some cases, the number of people involved in the same outbreak was not specified (Table 1).
46 strains vs. var. / variants / varietates vs. lineages ???? – This should be clarified. It is quite understandable that even the authors are not entirely clear on what term to use, and therefore use the term "strains" in quotation marks. However, it is worth considering whether it would not be more appropriate to use some other designation. The term strain would be appropriate for microorganisms (or asexually reproducing organisms) - typically, for example, for protozoa, bacteria, etc. However, in the case of mites, these are rather populations adapted to their hosts... (form / ecoform / ecotypes… ???)
Response. This terminology issue represents an ongoing debate among Sarcoptes scientists. Traditionally, the term “variant” is used to specify host-specific Sarcoptes scabiei (Arlian et al., 1989), although we might say that this subdivision is outdated and does not completely or always fit with the parasite epidemiology, but it is still accepted and widely used in the scientific literature.
In the revised version of the review, we decided to use the single term “variant”.
51 Anthropozoonosis – the definition is very ambiguous and differs from different sources; the understanding of this term is very different in different subdisciplines, and therefore it is better to avoid this word altogether and only use a descriptive form... that transmission from humans to animals has also been documented...
Response. We changed “anthropozoonosis” to “transmission from humans to animals”